# Human mobility patterns and malaria importation on Bioko Island

Carlos A. Guerra [1], Su Yun Kang[2], Daniel T. Citron [3], Dianna E.B. Hergott[4], Megan Perry[1], Jordan Smith[5], Wonder P. Phiri[5], José O. Osá Nfumu[5], Jeremías N. Mba Eyono[5], Katherine E. Battle [2], Harry S. Gibson[2], Guillermo A. García[1] & David L. Smith [3]

Malaria burden on Bioko Island has decreased significantly over the past 15 years. The impact of interventions on malaria prevalence, however, has recently stalled. Here, we use data from island-wide, annual malaria indicator surveys to investigate human movement patterns and their relationship to *Plasmodium falciparum* prevalence. Using geostatistical and mathematical modelling, we find that off-island travel is more prevalent in and around the capital, Malabo. The odds of malaria infection among off-island travelers are significantly higher than the rest of the population. We estimate that malaria importation rates are high enough to explain malaria prevalence in much of Malabo and its surroundings, and that local transmission is highest along the West Coast of the island. Despite uncertainty, these estimates of residual transmission and importation serve as a basis for evaluating progress towards elimination and for efficiently allocating resources as Bioko makes the transition from control to elimination.

---

[1] Medical Care Development International, 8401 Colesville Road, Suite 425, Silver Spring, MD 20910, USA. [2] Malaria Atlas Project, Big Data Institute, Nuffield Department of Medicine, University of Oxford, Roosevelt Drive, Oxford OX3 7FY, UK. [3] Institute for Health Metrics and Evaluation, University of Washington, 2301 Fifth Ave., Suite 600, Seattle, WA 98121, USA. [4] University of Washington, Department of Epidemiology, 1959 NE Pacific Street, Health Sciences Bldg, F-262, Box 357236, Seattle, WA 98195, USA. [5] Medical Care Development International, Avenida Parques de Africa S/N, Malabo, Equatorial Guinea. Correspondence and requests for materials should be addressed to C.A.G. (email: cguerra@mcd.org)

Malaria infection imported through human movement is of particular concern for countries and regions seeking elimination[1–10]. Imported malaria can precipitate local transmission in receptive areas and lead to persistence of the disease. Human hosts and mosquito vectors move about and with them the parasites that they carry. Human movement has a greater effect on malaria importation, however, because humans move at larger spatial scales and are able to transport parasites longer distances[1,3–9,11]. International travel accounts for much of the importation of parasites by humans returning from malaria endemic areas to non-endemic countries[12], but a very significant proportion of the human-related parasite movement takes place between regions of differing receptivity within an endemic country[6]. Understanding these movements is fundamental for local control and elimination efforts[2,3,9].

Located in the Gulf of Guinea, off the coast of Cameroon, Bioko is the main island of insular Equatorial Guinea (EG). It is administratively divided into four districts: Baney, Malabo, Riaba and Luba (Fig. 1). Its largest city, Malabo, is the country capital and home to around 85% of the population of the island. The Bioko Island Malaria Control Project (BIMCP) was established in 2003 with the aim to reduce the heavy burden of malaria[13]. Prior to this year, entomological inoculation rates (EIR) on Bioko were among the highest ever recorded for any malaria endemic area[14–16]. In 15 years of operation, the project has successfully reduced malaria transmission from hyper and holoendemic levels to largely hypoendemic and residual malaria chiefly through island-wide, intensive vector control and improved case management and prevention, though pockets of higher transmission intensity persist[17]. After the completion of the current phase by the end of 2018, strategies for the following five-year phase will aim at malaria elimination on the island[18]. While the specific plans are still being delineated, the inclusion of a pre-erythrocytic vaccine to the arsenal of control interventions is under consideration[19].

Imported malaria has been identified as a major challenge for malaria control and elimination on Bioko[20]. The island lies in the middle of a region where malaria transmission is mostly hyperendemic[21] and thus presents multiple sources of malaria parasites that can be carried during human travel (Fig. 1). In this study, we analyze infection prevalence and travel data assembled by the BIMCP as part of their annual malaria indicator surveys (MIS) to identify the patterns of human movement that can most significantly determine *Plasmodium falciparum* parasite importation. We also use maps of predicted travel prevalence (TP) and *P. falciparum* parasite rate (*Pf*PR), together with mathematical modeling, to provide preliminary estimates of the level of malaria importation through human travel. We find that the contribution of malaria infections acquired while travelling to local malaria prevalence on Bioko Island is significant and discuss its implications for the adoption of adequate and cost-effective malaria control strategies.

## Results

**Observed and predicted human travel.** A total of 17,016, 14,922 and 14,479 people were surveyed in 2015, 2016 and 2017, each sample representing more or less 6% of the population of the island. Overall, 20.2% of respondents had pernoctated at either another district or outside of Bioko at least one night within the preceding eight weeks; 12.2% had traveled to any destination off the island, 10.3% to Río Muni and 8.7% reported travel within Bioko (Table 1). When considering travel to within and off-island destinations together, no particular pattern was observed (Fig. 2). If they were considered separately, however, clear patterns were evident. Off-island travel, both to all destinations or specifically to Río Muni, was more common among inhabitants of Northern Bioko (*i.e.* Malabo and its surroundings; Fig. 1), whereas within-island travel was more commonly reported by people living along the East and West Coasts, and Southern Bioko. The geostatistical

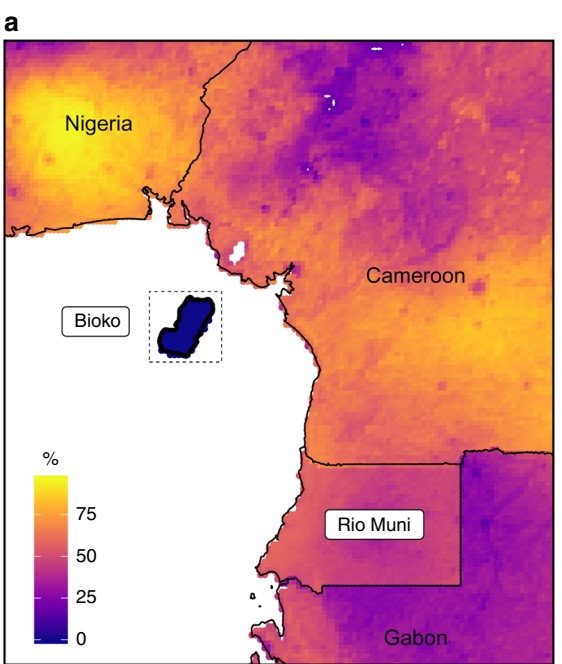

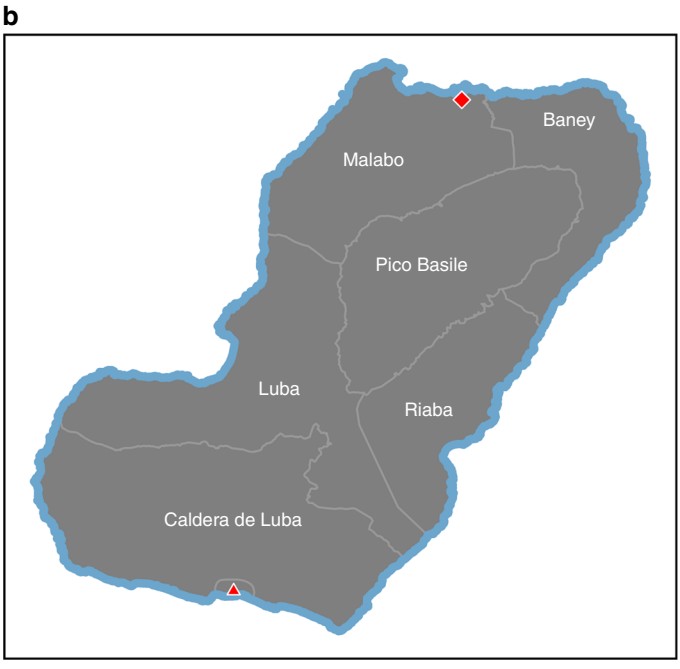

**Fig. 1** Bioko Island and its location in the Gulf of Guinea. **a** The continental territory of Equatorial Guinea is known as Río Muni. The color scale represents predicted *Pf*PR in children, reconstructed from data produced by the Malaria Atlas Project, which are available for use under the Creative Commons Attribution 3.0 Unported License[21,44]. **b** Detail of Bioko Island with its four districts (Malabo, Baney, Riaba and Luba) and two uninhabited nature reserves (Pico Basile National Park, to the North, and Caldera de Luba Scientific Reserve, to the South). The red diamond corresponds to the location of the capital city, Malabo, and the red triangle to that of the isolated town of Ureka, which is part of Luba district

**Table 1 Sample size and history of travel in respondents by each MIS**

| MIS year | 2015 | 2016 | 2017 | All years |
|---|---|---|---|---|
| Sample size | 17,016 (6.8) | 14,922 (5.9) | 14,479 (5.7) | 46,417 |
| Any travel (%) | 18.78 | 22.36 | 19.57 | 20.18 |
| Off-island travel, all destinations (%) | 12.23 | 13.17 | 11.02 | 12.15 |
| Off-island travel to Río Muni (%) | 10.80 | 10.48 | 9.37 | 10.25 |
| Within-island travel (%) | 7.06 | 9.90 | 9.21 | 8.65 |

Numbers within brackets next to the sample size are the approximate percentage of the human population of Bioko represented by the samples. Source data are provided as a Source Data file

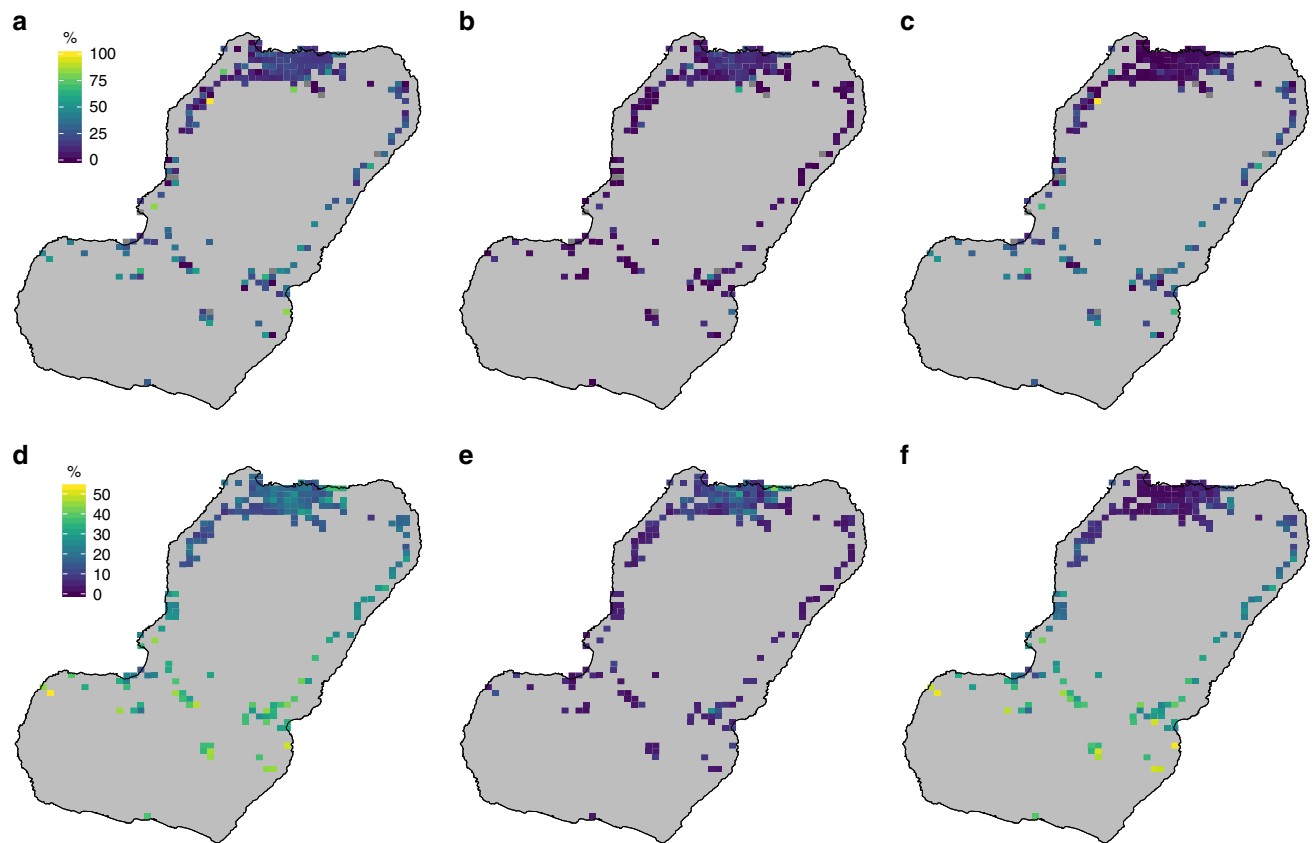

**Fig. 2** Travel prevalence on Bioko. The maps show the raw data (top row) and predicted surfaces (bottom row) as the percentage of people by map-area responding having traveled anywhere (**a**, **d**), to Río Muni (**b**, **e**) and within Bioko (**c**, **f**). Grey map-areas on the top row maps are those where no individuals were sampled in any of the three MIS due to low population. Note that the scale limits are different between the top and bottom rows. The noise introduced by sampling variance, illustrated by the extreme values of the top row, was smoothed out by the geostatistical models (bottom row). Source data are provided as a Source Data file

models effectively smoothed out sampling variance and provided a more coherent picture of TP (Fig. 2).

When analyzing within-island travel by district of origin and destination, a clear pattern was evident, whereby 89.8% of people inhabiting districts other than Malabo travelled to Malabo district. Conversely, domestic destinations of Malabo district residents were balanced mostly between Luba (44.3%) and Baney (43.6%; Table 2 and Fig. 3). The overwhelming majority of off-island travelers were bound to Río Muni (84.3%), 8.6% of them traveled to other African countries, 6.1% went to a destination outside of Africa, and the remaining 1% visited one of the other islands of EG.

**Malaria prevalence and travel history.** Table 3 presents the observed *Pf*PR in people with and without history of travel. *Pf*PR

for the three MIS weighted by sample size was 10.6% [95% CI 8.7–12.4%] in all individuals tested and 8.8% [95% CI: 6.9–10.7%] in individuals without history of travel. Conversely, *Pf*PR among people who traveled off the island was 23.3% [95% CI: 20.2–26.3%]. Travelers to Río Muni had a *Pf*PR of 26.6% [95% CI: 23.1–30.2%], in contrast with travelers to other off-island destinations in whom *Pf*PR ranged between 5.4 and 7.1%. Within-island travelers showed an overall malaria prevalence of 6.6% [95% CI: 4.0–9.3%]. *Pf*PR in off-island travelers was significantly higher than *Pf*PR in all individuals and non travelers ($p < 0.01$ Wilcoxon rank sum test; Fig. 4). The data showed increased odds of malaria infection in off-island travelers compared to non-travelers (OR: 3.0 [95% CI: 2.8–3.2]), especially when considering travel to Río Muni (OR: 3.6 [95% CI: 3.4–3.9]; Table 3). We used the median value of the proportion of off-island travelers (5.3%) to classify map-areas into low and high travel and found that,

**Table 2 Number of people reporting within-island travel according to district of residence and district of destination**

| District | Baney | Malabo | Luba | Riaba | All non-Malabo |
|---|---|---|---|---|---|
| Baney | — | 568 (43.6) | 22 (2.2) | 29 (6.9) | 51 (2.2) |
| Malabo | 688 (82.9) | — | 971 (97.2) | 360 (85.9) | 2019 (89.8) |
| Luba | 104 (12.5) | 576 (44.3) | — | 30 (7.2) | 134 (6.0) |
| Riaba | 38 (4.6) | 158 (12.1) | 6 (0.6) | — | 44 (2.0) |
| Total | 830 (100) | 1302 (100) | 999 (100) | 419 (100) | 2248 (100) |

Figures within brackets are the percentages of the totals. The destination districts are listed in the first column. The column headers name the district of origin. Source data are provided as a Source Data file

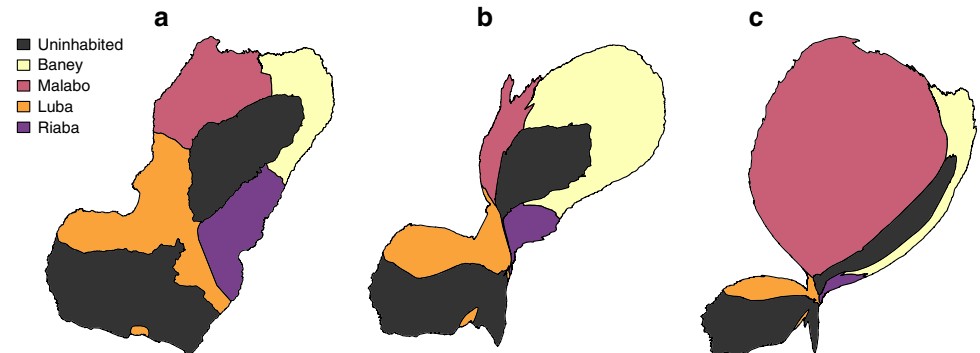

**Fig. 3** Cartograms of Bioko Island illustrating within-island destinations. Areas were distorted in proportion to the number of within-island travelers bound to each destination district (Table 2). **a** The administrative units of Bioko Island are shown with their correct shape and size, for comparison. **b**, **c** The destinations of those who traveled from Malabo and of those traveling from all other districts are illustrated as cartograms, respectively. Each cartogram was produced using 10 iterations. Source data are provided as a Source Data file

**Table 3 Map-area level malaria prevalence data**

| MIS year | 2015 | 2016 | 2017 | All |
|---|---|---|---|---|
| Sample size | 109.8 [76.1–143.4] | 86.6 [60.2–112.9] | 83.1 [57.8–108.4] | 83.2 [57.8–108.7] |
| *Pf*PR in all tested | 12.7 [10.0–15.5] | 8.5 [7.0–10.1] | 10.2 [7.4–12.9] | 10.6 [8.7–12.4] |
| *Pf*PR in non travelers | 10.8 [8.2–13.5] | 6.8 [5.2–8.2] | 8.4 [5.6–11.3] | 8.8 [6.9–10.7] |
| *Pf*PR in off-island travelers | 25.9 [20.2–31.6] | 19.7 [14.6–24.8] | 24.2 [19.6–28.9] | 23.3 [20.2–26.3] |
| *Pf*PR in travelers to Río Muni | 28.8 [22.9–34.7] | 23.4 [17.9–28.9] | 27.5 [22.2–32.8] | 26.6 [23.1–30.2] |
| *Pf*PR in travelers to other EG islands | NA | 6.8 [0–18.5] | 7.7 [0–28.1] | 7.1 [0.2–14.1] |
| *Pf*PR in travelers to other African destinations | 3.8 [1.8–5.8] | 7.0 [1.2–12.9] | 4.8 [1.5–8.2] | 5.4 [2.2–8.6] |
| *Pf*PR in travelers to destinations out of Africa | 6.3 [0–15.5] | 1.5 [0.7–2.3] | 9.5 [2.6–16.3] | 5.4 [0–10.9] |
| *Pf*PR in within-island travelers | 8.1 [4.7–11.5] | 5.2 [2.8–7.5] | 7.0 [3.4–10.6] | 6.6 [4.0–9.3] |
| OR for off-island travel, all destinations | 2.9 [2.6–3.2] | 3.4 [2.9–3.8] | 3.4 [3.0–3.9] | 3.0 [2.8–3.2] |
| OR for off-island travel to Río Muni | 3.3 [3.0–3.7] | 4.2 [3.7–4.8] | 4.1 [3.6–4.7] | 3.6 [3.3–3.9] |

*Pf*PR values are in percent. All figures, except for odds ratios (OR), are sample size weighted means and numbers within brackets indicate 95% confidence intervals. Source data are provided as a Source Data file

after excluding travelers, the median *Pf*PR in the former was significantly lower (*p* < 0.01 Wilcoxon rank sum test; Fig. 4).

The predicted malaria prevalence surfaces confirmed the patterns revealed by the raw data, with higher *Pf*PR towards the West and North of Bioko; this pattern was similar for *Pf*PR in all individuals (*Pf*PR$_{all}$) and *Pf*PR in non-travelers (*Pf*PR$_{nt}$). When looking at *Pf*PR in travelers to Río Muni (*Pf*PR$_{rm}$), however, higher prevalence was evident throughout, including in and around Malabo (Fig. 5). The ratio of the mean *Pf*PR$_{rm}$ to the mean *Pf*PR$_{nt}$ highlighted those areas where the infection among travelers probably contributed more to the overall *Pf*PR than local transmission, and this was the case for the great majority of map-areas (179 out of 194, 92.3%; Fig. 6).

**Local malaria transmission and importation.** Our findings showed that *Pf*PR was significantly higher in those who had traveled to mainland EG than *Pf*PR in all individuals, yet in individuals with history of within-island travel it was actually lower. We therefore sought to derive an estimate for the total fraction of all observed infections that were attributable to exposure in mainland EG. Figure 6 plots the travel fraction and local residual transmission estimates by map-area based on the raw data and on the predicted surfaces. The former suffered from noisy data and several missing values. Supplementary Figures 1 and 2 present the upper and lower bounds for the latter estimates. The travel fraction was estimated at 100% in much of Malabo and in a few isolated populations to the East and South of Bioko. An

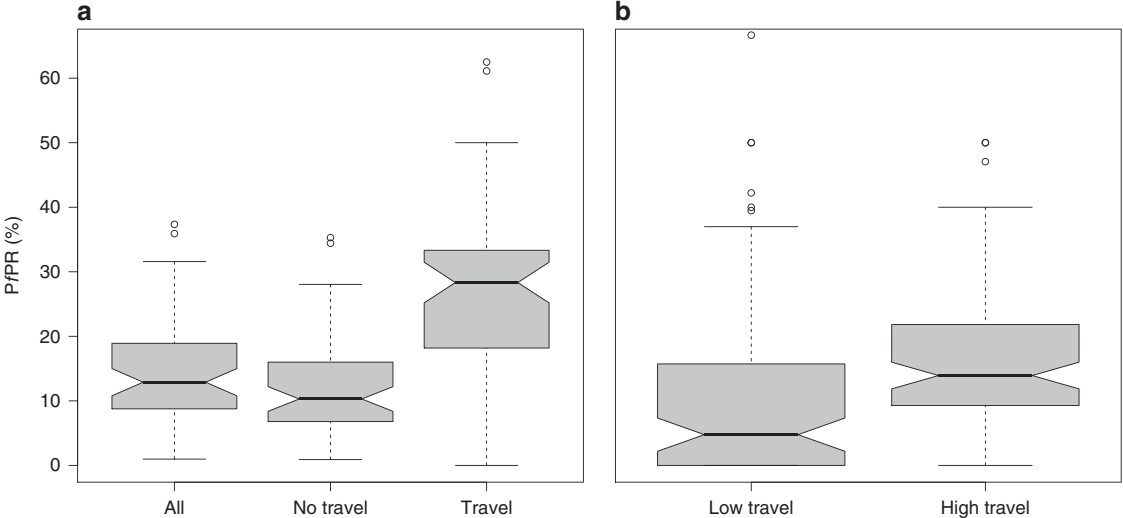

**Fig. 4** Distribution of observed *Pf*PR across map-areas according to history of travel. **a** *Pf*PR in all individuals and in those with and without history of off-island travel. **b** Distribution of *Pf*PR across map-areas with low and high levels of off-island travel, after excluding travelers. The cutoff was determined by the median travel prevalence for all map-areas (see text). The thick black lines and notches in the boxes represent the medians and associated 95% confidence intervals; the box limits represent the interquartile range; the minimum and maximum values are given by the horizontal lines at the end of the whiskers and outliers, or data points that are lower or greater than the first and third quartile by 1.5 times the interquartile range, are represented by the empty circles. Source data are provided as a Source Data file

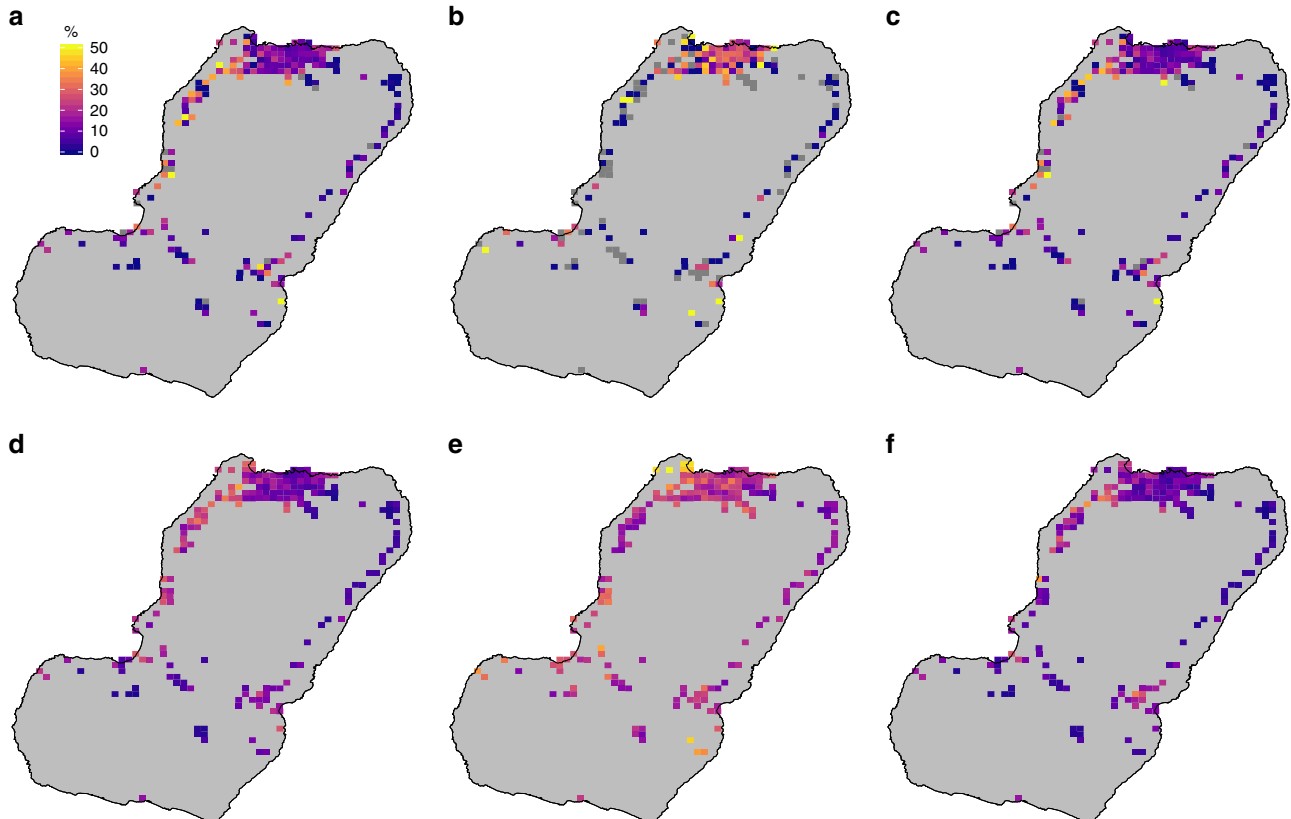

**Fig. 5** Observed (top row) and mean predicted *Pf*PR (bottom row). **a**, **d** All individuals. **b**, **e** Off-island travelers to Río Muni. **c**, **f** Non-travelers. Grey map-areas in (**a**) are those where no individuals were sampled in any of the three MIS due to low population. Grey map-areas in (**b**, **c**) also indicate places where no individuals responded to travel history and hence *Pf*PR in travelers and non-travelers could not be estimated. The noise introduced by sampling variance, illustrated by the extreme values of the top row, was smoothed out by the geostatistical models (bottom row). Source data are provided as a Source Data file

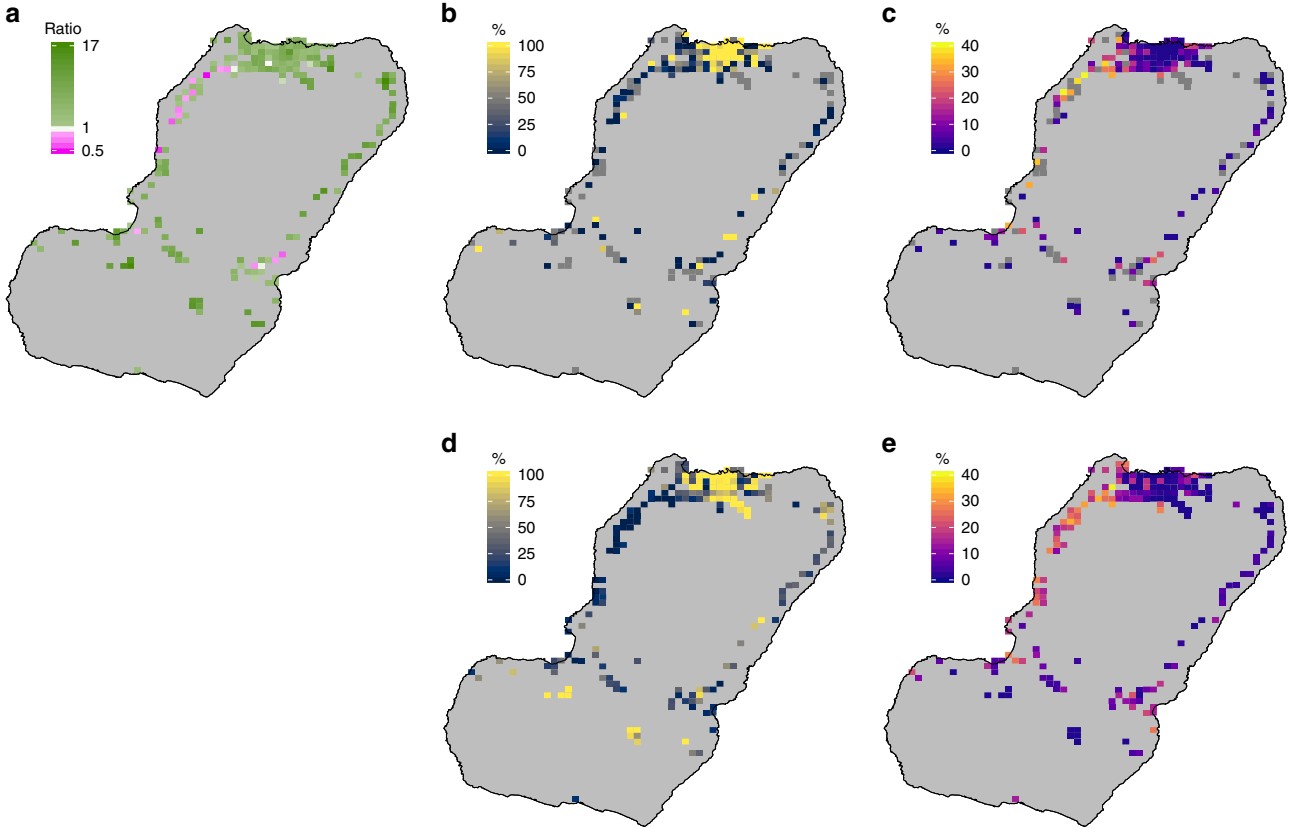

**Fig. 6** Estimating the degree of malaria importation. **a** Ratio of the mean predicted $PfPR_{rm}$ to the mean predicted $PfPR_{nt}$. Magenta pixels are map-areas where predicted $PfPR_{nt}$ is higher, green ones where predicted $PfPR_{rm}$ is higher and white pixels where both $PfPR$ are the same. **b**, **c** depict, respectively, the mean travel fraction of malaria prevalence and the local residual malaria estimated by our model based on the raw data. **d**, **e** are the same estimates but based on the geostatistical surfaces. In **b**, **d**, light, yellow pixels suggest that malaria in those map-areas is imported and hence local residual malaria could be zero. In the absence of malaria prevalence attributable to travel, the residual malaria is estimated to be highest mostly in the Northwest of Bioko (lighter, orange to yellow pixels in **c**, **e**). Source data are provided as a Source Data file

estimated 48.4% [lower bound 9.8%; upper bound 58.3%] of the population of Bioko could live in map-areas with no local transmission. Accordingly, 67.2% [lower bound 27.2%; upper bound 70.9%] and 74.8% [lower bound 62.7%; upper bound 83.5%] could live in map-areas where the travel fraction was estimated at 80% or higher and 50% or higher, respectively (see Supplementary Table 1). Our analyses returned an estimated overall population-weighted mean prevalence due to local residual transmission of 5.1%, with higher estimated prevalence mostly along the West Coast, and in a few isolated areas of Malabo and Riaba (Fig. 6). In some of these areas, the prevalence of malaria would likely be higher even in the absence of malaria importation. Conversely, according to our model local transmission in Malabo contributes little to the malaria prevalence measured there.

## Discussion

In its fourth phase beginning in 2019, the BIMCP has redefined itself into the Bioko Island Malaria Elimination Project (BIMEP) to integrate new intervention strategies aiming to rid the island of malaria[22]. Even though substantial gains have been achieved, there is a long path ahead before reaching the ultimate goal of elimination. One of the main challenges is tackling malaria importation. This study attempted to better quantify the extent of the problem of imported malaria and explain how human travel might shape this phenomenon. Our findings show clear human mobility patterns operating on the island and characterizing them

is a stepping stone towards understanding malaria importation. Crucially, our results suggest that the degree of malaria importation, and its contribution to malaria prevalence on Bioko, probably has been highly underestimated.

Around a fifth of the human subjects surveyed in the last three annual MIS had travelled at least once, within eight weeks prior to the survey, to any destination outside of their district of residence. Anecdotal reports suggest that the human population of Bioko is highly mobile and travels frequently from rural areas to Malabo for employment and schooling. We observed a pattern for within-island travel whereby between 83 and 97% of non-residents of Malabo traveled to the city from other districts (Table 2), a pattern resembling gravity models that explain population movements as a function of distance and population size[23,24]. In effect, the choice of distance to Malabo and population density as covariates for the travel surfaces was based on these findings. The higher prevalence of travel to mainland EG by Malabo residents does not follow the same gravity assumption, however. The city of Bata, in Río Muni, is the main port of entry for inbound flights and boats from Malabo, but the population of Bata is similar in size to that of Malabo and the travel time between both cities, a proxy of distance, differs substantially according to the means of transportation (around 45 min by air and 15 h by boat, on average). It is possible that people that live in Malabo can afford more trips abroad because they are generally wealthier than the rest of the population and this could partly explain the off-island travel pattern observed[25–27]. Another factor could be that, because the main government offices are based in Malabo and

Bata, public servants move frequently between Bioko and the mainland. The demographics of Bioko and Río Muni may also play a role; Fang people predominate in Río Muni and also represent a large proportion of the population of Malabo. Conversely, the South of Bioko is predominantly inhabited by Bubi people, who are relatively few in Río Muni. These demographic differences may influence travel connection patterns.

Our findings suggest that malaria cases are actively imported by off-island travelers to Río Muni, similar to what has been described in Zanzibar[1,4]. The MIS data revealed that overall PfPR was significantly higher in areas where travel to mainland EG was more prevalent and that the odds of malaria infection in these travelers were more than three times the rest of the population (Fig. 4; Table 3). Moreover, our modeled estimates indicated that around half of the population of Bioko lived in areas where the travel fraction could reach 100% and that this phenomenon was particularly apparent in Malabo. This does not mean that there is no transmission occurring in these areas but rather that it is plausible that travel could account for all or most of the malaria observed. In fact, anopheline vectors have been collected in Malabo, albeit at densities that are an order of magnitude lower than at entomological surveillance sites elsewhere on the island, and these vectors may sustain local transmission in the city. On the other hand, our estimates could partly explain the atypical malaria prevalence found in Malabo, which is unusually high for an urban area[28]. This is not surprising given that PfPR in children in Río Muni ranges between 32 and 59%[21,29] (Fig. 1), corresponding to mesoendemic and hyperendemic malaria transmission. During a boat survey conducted in 2013, Bradley et al. estimated a PfPR of 70.4% in children traveling from mainland to Bioko in contrast with 38.1% prevalence in young passengers traveling in the opposite direction[20]. During the same boat survey it was revealed that 86% of Bioko residents traveling to Río Muni intended to stay longer than a week. A similar finding was true for mainland residents visiting Bioko, with 85% of them intending to stay for more than one week on the island[20]. Moreover, the duration of stay of around a quarter of travelers in both directions was reportedly longer than seven weeks. The longer that people stay at destinations where malaria transmission is high, the higher the chances of acquiring an infection; someone who spends a couple of days is less likely to import new infections than someone who spends a couple of weeks.

The probability of onward transmission after the importation of cases depends on the local receptivity of areas acting as parasite sinks, which is largely dependent on local vectors and vector control[8,9]. Despite the significant achievements of the BIMCP, including the elimination of the formerly dominant vector Anopheles funestus[30,31] during the early stages of implementation[14] and dramatic reductions in EIR shown in recent surveys[18], resilient populations of An. coluzzi and An. melas persist. These populations were responsible for maintaining a mean EIR of 13 infected bites per person per year measured at sentinel sites in 2017[18]. Unfortunately, given the limited number of such estimates and the disproportionately smaller number of sites located in urban Malabo, these data proved insufficient to cross-validate our estimates of local residual transmission. Local receptivity to malaria is a problem in those areas where vector populations thrive despite intensive control with insecticide residual spraying and mass distribution of insecticide treated mosquito nets. Although a relationship between within-island human travel and malaria prevalence was not apparent, it is intuitive to assume that it probably plays an important role in mixing imported parasites from mainland as locals travel back from Malabo to rural communities where mosquito biting rates are still high.

The main caveats of our analyses are given by the limitations of the data. First, the destinations of travelers were only broadly recorded and hinder an accurate assessment of the vulnerability linked to human travel. By recording more specifically the destination of travelers to mainland EG it should be possible to determine the demographic sources of parasites with a higher degree of spatial certainty using available resources[9,21], to better assess risk of malaria infection in travelers and, therefore, to discriminate imported malaria from locally acquired cases. In a similar fashion, knowing with better precision the destinations of within-island travel would help to understand the dynamics of onward transmission of imported parasites. Distances are relatively short on Bioko and people move around frequently between areas with differing receptivity during the peak biting times of anopheline vectors. This micro-scale movement was impossible to characterize from the MIS data. Although people tend to spend the great majority of their time at home, even small amounts of time exposed to mosquito bites can lead to infection[10]. Second, human movement relevant for malaria epidemiology is seasonal[32]. In areas that have successfully brought transmission down but that remain receptive to malaria, transmission may become seasonal following human movement and this seasonality is an important consideration for planning malaria elimination strategies. Searle et al. showed that human movement in southern Zambia was affected by seasonality and that this had an effect on local malaria epidemiology[10]. Here we used three cross-sectional snapshots of malaria prevalence and human travel comprising the eight weeks prior to each MIS. It was not possible to infer frequency of travel or to capture any seasonal component of either human infection or movement patterns. Third, the MIS sampling is sometimes biased by the absence of working adult men during the surveys; since this cohort of the population may have different travel habits, this introduced an unknown bias into our estimates. Finally, the data provided insights of the movement of returning residents that act as passive acquirers to bring parasites back to Bioko. To fully estimate importation rates, visitors to Bioko that carry parasites from mainland sources and act as active transmitters also must be accounted for[6], although their contribution to overall importation probably would be comparatively small[1].

We addressed these caveats partly by adjusting the assumptions of our model and testing them through sensitivity analyses (see Supplementary Notes 6 to 8, Supplementary Table 2 and Supplementary Figs. 3 to 5). Importantly, our findings prompted the modification of the questionnaire for the 2018 MIS to more explicitly capture some of this missing information, including the specific destinations of both off-island and within-island travel, the length of stay during travel and the frequency of travel within a specified period (i.e. how many times respondents traveled as opposed to only recording whether they had traveled or not within the specified time frame). Therefore, the new MIS will provide better data for future analyses. In addition, new, more comprehensive surveys of both boat and air passengers would be desirable and could help to further understand traveling patterns between the mainland and Bioko and better quantify malaria importation rates. These surveys, however, are both logistically difficult and resource intensive making them possibly unfeasible in practice.

As Bioko approaches pre-elimination, addressing the problem of malaria importation becomes increasingly relevant. Because much of the PfPR currently observed on the island could probably be accounted for by imported cases, novel control strategies designed to address specifically case importation should be considered[33,34]. Targeted malaria control in high-risk areas of mainland EG where travelers go most would reduce the main source of imported parasites to Bioko. In addition, interventions

**Table 4 List of environmental and socio-demographic covariates used for predicting *Pf*PR and TP**

| Covariate | Description | Dynamic | Source |
|---|---|---|---|
| Accessibility | Distance to cities with populations >50,000 | Static | Nelson et al.[45] |
| AI | Aridity index | Static | World Clim[46] |
| DistToWater | GIS derived surface that measures distance to permanent and semi-permanent water based on presence of lakes, wetlands, rivers, streams and accounting for slope and precipitation | Static | MAP (from WWF surfaces[47, 48]) |
| Elevation | Elevation as measured by the shuttle radar topography mission (SRTM) | Static | SRTM derivative[49] |
| Landcover | Types of land cover including cropland, forest, grass savanna, shrubland, urban/barren, water, wetlands, and woody savanna | Static | MODIS derivative[50] |
| PET | Potential evapotranspiration | Static | World Clim[46] |
| Slope | GIS derived surface calculated from SRTM elevation surface | Static | MAP (from SRTM[49]) |
| Stable_lights | Index that measures the presence of lights from towns, cities and other sites with persistent lighting | Static | NOAA[51] |
| TWI | Topographic wetness index | Static | MAP (from SRTM[49]) |
| Distance to Malabo[a] | Distance to central Malabo | Static | This paper |
| Population size[b] | Estimated population per 1 × 1 km pixel | Annual | WorldPop[52] |
| EVI | Enhanced vegetation index | Annual | MODIS derivative[53] |
| LST_day | Day time land surface temperature | Annual | MODIS derivative[54] |
| LST_delta | Diurnal difference in land surface temperature | Annual | MODIS derivative[54] |
| LST_night | Night time land surface temperature | Annual | MODIS derivative[54] |
| TCB | Tasseled Cap Brightness; measure of land reflectance | Annual | MODIS derivative[55] |
| TCW | Tasseled Cap Wetness | Annual | MODIS derivative[55] |
| TSI | Temperature suitability index | Annual | MAP[56] |

[a]Used for the TP surfaces only
[b]Used for both the TP and *Pf*PR surfaces

such as malaria screening at ports and protective measures specifically targeted at high-risk travelers may prove onerous to implement but would help to directly curb the impact of imported malaria. Ongoing work is developing more sophisticated models to simulate how a vaccine could help eliminate malaria under different scenarios of efficacy and coverage[19], considering also various levels of malaria importation. These new modeling efforts will incorporate travel models that capture the human movement patterns described in this paper and will also take advantage of better data from more detailed MIS questionnaires. Despite the progress, substantially more work is needed to improve our understanding of malaria importation in order to prepare Bioko for elimination.

## Methods

**Data.** For operational purposes, the BIMCP divides Bioko into 2,091 uniquely coded, 1 × 1 km map-areas, 194 of which were inhabited according to a 2015 population census[35]. Since the start of the BIMCP, individual level, geo-positioned data have been collected every year through annual MIS. We used data from the last three years available: 2015, 2016 and 2017[25–27] and aggregated them at the map-area level. The surveys were conducted in August and September each year, during the peak of the main rainy season. Nine map-areas were not surveyed in any of the three MIS due to low human population (*i.e.* data were available for 185 of the 194 inhabited map-areas). We focused our analyses on two types of data: malaria prevalence and travel data. For the purposes of measuring malaria, we included data from individuals of all ages because we found no significant differences in *Pf*PR between the overall sample population and children (2–10 year olds). During the MIS, all individuals present in their household during each survey were tested for the presence of *P. falciparum* in their blood using Carestart (Access Bio) rapid diagnostic tests (RDTs) that we used for deriving a *Pf*PR for each map-area. People were also asked whether they had traveled outside of their district or outside of Bioko within eight weeks prior to the survey and stayed away for at least one night. We defined TP per map-area as the percentage of people in each sampling unit answering yes to within island travel, off-island travel and both. Within and off-island travel questions were treated separately in the questionnaires; that is to say, the same individual could have answered positively to history of travel off the island and within the island. It was not possible to assess frequency of travel with accuracy because the questionnaire responses were binned into three categories (1–3, 4–10 and >10 travel occurrences within the stipulated period) with most people (97–99%, depending on the survey year) falling in the first category. The destinations of these travels were recorded broadly. For within-island travel, the final destination was recorded at the district level (Fig. 1). For off-island destinations, the answers were constrained to: Río Muni (*i.e.* mainland EG), other islands of insular EG (mainly, Annobón and Corisco), other African countries, and any

destination outside of Africa. Unless otherwise stated, we focused our analyses on off-island travel destined to mainland EG.

**Predicting travel and *Pf*PR.** Even though individuals were surveyed in most map-areas, in some, sample size variance affected the credibility of *Pf*PR and TP estimates derived from the data. Moreover, there were several map-areas where there were no off-island travelers and hence no denominator for estimating *Pf*PR in travelers. We therefore resorted to using the geo-positioned *Pf*PR and TP raw estimates as inputs for a Bayesian geostatistical modeling framework[36]. This methodology allowed not only to estimate *Pf*PR and TP in pixels where no individuals had been sampled in order to produce continuous surfaces for each variable but also to account for the sample size variance across all map-areas. For *Pf*PR, environmental and socio-demographic variables known to interact with and influence PfPR were assembled as 30 arcsecond spatial grids. For predicting TP, the covariates used were population density and distance to Malabo (Table 4). These variables were also input data for the Bayesian geostatistical model.

*Pf*PR in all individuals, in off-island travelers and in within-island travelers, as well as TP for any travel, off-island travel and within island travel were modeled via a Bayesian binomial logistic regression model with spatial random effects accounting for a spatial latent process. The integrated nested Laplace approximation (INLA) approach[37] was adopted for model inference and prediction via the R-INLA package[38]. Model description is as follows.

Let $Y_s$, $n_s$, and $p_s$ be the number of positive individuals (*i.e.* either malaria infected or recent traveler), the number of individuals screened, and prevalence (*Pf*PR or TP) at geo-coded location $s$ ($s = 1, \ldots, N$). $Y_s$ is assumed to follow a binomial distribution:

$$Y_s \sim \text{Bin}(p_s, n_s). \tag{1}$$

The prevalence of infection (or travel), $p_s$, is modeled via a linear regression on the logit scale:

$$\text{logit}(p_s) = X_s^T \beta + \phi_s. \tag{2}$$

The matrix $X$ includes an intercept and a list of environmental and socio-demographic covariates known to affect *Pf*PR. For the travel surfaces, the matrix included only distance to Malabo and population density. $\beta$ is the vector of regression coefficient, and $\phi_s$ is the continuously-indexed Gaussian random field. The Gaussian random field was modeled using stochastic partial differential equations which represent a Matérn spatial Gaussian field as a Gaussian Markov random field via triangulation[39].

The variable selection procedure started with examining collinearity among the environmental covariates by calculating variance inflation factors (VIF) prior to fitting the statistical model to the data. A stepwise selection of covariates using VIF was undertaken to make sure all VIF values were below a desired threshold (VIF < 10 in this case). A VIF threshold of 10 was chosen based on the commonly recommended rule of thumb for reducing collinearity[40]. Besides, we found that a smaller or larger VIF threshold value did not influence the outcome of covariate selection. Using the full set of covariates, a VIF for each variable was calculated, the

variable with the single highest value was removed, all VIF values with the new set of variables were recalculated, the variable with the next highest value was removed, and so on, until all VIF values were below the threshold of 10. Subsequently, variable selection was performed using bidirectional elimination by running stepwise regressions on all model combinations using the shortlisted covariates and calculating the deviance information criterion (DIC) for each model. The final model was the one with the smallest DIC value. We note that reducing the VIF was associated with reducing the degree of multi-collinearity of the covariates in a regression model and it certainly did not imply a reduction in DIC. The VIF reduction step was done prior to fitting the Bayesian geostatistical model to the data. On the other hand, the DIC reduction step was part of the Bayesian geostatistical model fitting and selection procedures for identifying the final and most parsimonious model that was then used to predict prevalence rates across Bioko Island. The DIC has been widely justified and recommended as a standard criterion for comparing complex Bayesian hierarchical models[41]. Using the final model, $Pf$PR and TP were predicted over a 30 arcsecond (i.e., approximately $1 \times 1$ km spatial resolution) spatial grid of the 2,091 pixels that cover Bioko island. The input data for $Pf$PR were divided into four different sets: all the population, those with history of travel to any destination off the island, those with history of travel to Río Muni and those without history of off-island travel. For the TP predictions, four different input data sets were used as well: all travel, off-island travel to any destination, off-island travel to Río Muni and within-island travel. The resulting predictions were depicted as maps of mean $Pf$PR and mean TP for each of these sets. A range of model validation analyses were used to assess the model's goodness-of-fit and predictive accuracy, including Pearson correlation of observed and predicted data, and validation runs with incremental hold-out validation data subsets. The credible intervals for the covariates and geo-statistical model outputs are presented in Supplementary Note 9, Supplementary Tables 3 to 8 and Supplementary Fig. 6.

**Estimating malaria importation.** Using the predicted surfaces, we calculated the ratio of the mean predicted $Pf$PR in off-island travelers to Río Muni ($Pf$PR$_{rm}$) to the mean predicted $Pf$PR in non-travelers ($Pf$PR$_{nt}$) to gauge the relative importance of imported malaria. We also developed a model to estimate the travel fraction, or the fraction of the malaria positive population attributable to travel. The population that was malaria positive could include people who had traveled within the eight week period, but also many who could have traveled before the eight weeks but remained infected. As input data for this simple mathematical model we used the predicted surfaces of $Pf$PR in all individuals ($Pf$PR$_{all}$), $Pf$PR$_{rm}$ and travel prevalence to Río Muni (TP$_{rm}$).

Let $h$ denote the daily force of infection, and $\delta$ the daily off-island travel rate (i.e. the rate at which people return from Río Muni). Let $\eta$ denote the portion who acquired malaria while traveling off island. The term $\eta\delta$ is like $h$, equivalent to a force of infection from off-island travel. Let $r$ denote the daily decay rate of malaria prevalence in the absence of treatment, where $r^{-1} \approx 200$ days[42]. The mathematical model for the overall prevalence of infection, PR, is:

$$\frac{dPR}{dt} = (h + \eta\delta)(1 - PR) - rPR, \tag{3}$$

where at the steady state:

$$PR = \frac{h + \eta\delta}{h + \eta\delta + r}. \tag{4}$$

In the absence of local transmission, prevalence would be entirely attributable to travel, PR$_{T0}$:

$$PR_{T0} = \frac{\eta\delta}{\eta\delta + r}. \tag{5}$$

We defined the travel fraction as TF $=$ PR$_{T0}$/PR. Conversely, in the absence of imported malaria, prevalence from local residual transmission, PR$_{L0}$, would be:

$$PR_{L0} = \frac{h}{h + r} \tag{6}$$

Using the model, we estimated $\delta$, $\eta$ and $h$ from TP$_{rm}$ and $Pf$PR$_{rm}$. As an upper bound, we assumed that everyone who returned with malaria had acquired it while traveling in Río Muni. As a lower bound, we assumed that travelers who left Bioko bound to Río Muni were already infected with the same probability as the average population of that particular map-area. In between these bounds we assumed that parting travelers were already infected due to local residual transmission; these last estimates, which are the ones shown in the Results, involved co-estimating $h$ and $\eta$ in order to estimate for local residual transmission and travel fraction. We used human population data from a recent census[35] to estimate the percentage of inhabitants living in areas of different estimated travel fractions. We also conducted sensitivity analyses on the duration of infection ($r$), heterogeneity in travel, $h$ and $\eta$. See Supplementary Notes 1 to 5 for a full explanation of our assumptions and mathematical derivations.

All analyses were performed and all figures created using R 3.5.3[43].

**Reporting summary.** Further information on research design is available in the Nature Research Reporting Summary linked to this article.

## Data availability

Source data for the $Pf$PR estimates in Figure 1 are available from the Malaria Atlas Project (MAP) at https://map.ox.ac.uk/. Source data for Figs. 2 to 5 and Tables 1 to 3 are provided with the paper as a Source Data file and are also available at https://doi.org/10.6084/m9.figshare.8009684.v1. These are raw data aggregated at map-area level and selected data from the individual-level MIS databases. The full individual-level data are not publicly available due to them containing information that could compromise privacy/consent of surveyed individuals.

## Code availability

The mathematical code developed to estimate local residual transmission and travel fraction is available in R language from https://doi.org/10.6084/m9.figshare.8009684.v1. The input data for this code are provided within the Source Data file to produce Fig. 6 and all Supplementary Information figures.

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

## Acknowledgements

D.L.S., S.Y.K. and D.T.C. acknowledge support from a grant from the Bill & Melinda Gates Foundation OPP1110495. We are thankful to Sean L. Wu, Héctor M. Sánchez-Castellanos, Alec Georgoff, John Henry and Qian Zhang for fruitful conversations and general advice on the analyses and models. We thank the participants on Bioko Island who have taken part in the surveys during the study period, and the BIMCP field surveyors and supervisors who collected the data. We also acknowledge the BIMCP mapping team who have sustained the geographic tracking system that made these analyses possible. Finally, we acknowledge the National Malaria Control Program and the Ministry of Health and Social Welfare of Equatorial Guinea, as well as Marathon Oil, Noble Energy, AMPCO (Atlantic Methanol Production Company) and the Ministry of Mines and Energy of Equatorial Guinea for their continued efforts and support of malaria control on Bioko Island.

## Author contributions

C.A.G., D.L.S. and D.T.C. conceived the analyses. C.A.G. processed and analyzed the MIS data. D.L.S. and D.T.C. developed the mathematical model and undertook the sensitivity analyses. S.Y.K. and C.A.G. adapted and ran the geostatistical models and wrote the relevant section of the Methods. D.T.C. and D.L.S. wrote the Supplementary Information. K.E.B. and H.S.G. assembled the covariate layers. D.E.B.H., M.P., J.S., W.P.P., J.O.O. N., J.N. and G.A.G. collected the MIS data and assembled the databases. C.A.G. and D.L. S. wrote the manuscript. All authors commented and approved the final manuscript.

## Additional information

**Competing interests:** The authors declare no competing interests.

