## [Peer Review File · Nature Communications]

Reviewers' comments:

Reviewer #1 (Remarks to the Author):

"Human mobility patterns and malaria importation on Bioko Island" is a really interesting paper and a very good case study since it is carried out in a close system (an island).

The manuscript is well written and the mathematical and statistical models well described. It lacks in showing the full results (i.e. uncertainties, variable selection, and parameterisation) and in the description of some choices and assumptions. In my view, this paper is suitable for Nature Communications but still needs essential improvements.

Major points to be addressed.

1) My major concern is related to how the data is treated. It is not clear if the authors have considered and how they considered multiple travels, i.e. same people doing within- and off-island travels in the same period. I can't see anything in the statistical and mathematical models accounting for joint (within and off in they different flavours) travels. Until this issue is not addressed, statements such as "our findings show clear human mobility patterns operating on the island and characterising them is a stepping stone towards understanding malaria importation" or "our findings suggest that malaria cases are actively imported by off-island travellers to Rio Muni, similar to what has been described in Zanzibar" are potentially highly speculative.

2) Please add a table with the variable selection results and explain why you choose a VIF threshold of 10.

3) Reducing DIC and VIF are not the same thing, and actually reducing VIF does not imply a reduction in DIC. Can you provide more details and your assumptions about model reduction and best fitting please?

4) Include tables with the Bayesian geostatistical models results, i.e. credible intervals for parameters; and the uncertainty maps.

Minor points

5) In the background section, "historically, entomological inoculation rate...", please can you be more specific about the time frame considered in the word "historically".

6) In the background section, "the project will enter a new era of malaria elimination...", although the wording is very exciting it does not help me to understand what is this "new era" in malaria elimination.

7) Figure 1. Include the label for the Pico Basile National Park and the Caldera de Luba Scientific research.

8) In the results section, please can you add the proportion of total population represented by the sample size (i.e. for 2015, 17016 people are % of total population).

9) "because we did not find a significant difference between PfPr in children and adults..." I think this need to be discussed in more details, does it mean that they have similar travel and malaria PfPR patterns? If yes why?

Reviewer #2 (Remarks to the Author):

I enjoyed reviewing this paper, the key finding of which is that malaria prevalence in much of Bioko (particularly the main city) is consistent with being largely due to importation from the mainland, while malaria outside the main city cannot be explained by importation alone. This is an important conclusion that has critical implications for designing appropriate elimination programs in Bioko – i.e., not all places with high prevalence should necessarily be targeted with vector control measures if transmission is in fact happening elsewhere - and the methods used for deriving it are novel and potentially of use to many other countries facing similar challenges.

My chief comments and criticisms are:

1) The population surveys provided information on malaria testing and travel patterns in 185/194 map areas. Geostatistical modeling is used in this analysis to derive estimates for the entire

population. These modeled extrapolations seem useful to explain, for example, what percent of the total population lives in areas defined by certain endemicity or travel characteristics, accounting for any gaps in sampling. However, it is not clear to me why the survey data could not be directly analyzed instead of using the geostatistical model results to investigate the degree to which importation drives prevalence. My concern here is that the modelling procedure is likely to smooth the data, potentially averaging over interesting heterogeneities that could hold importance for the questions posed here (i.e., pixels home to clusters of unusually high or low travel or malaria may have important roles in driving transmission). Would results change if survey-measured values were used directly to inform the model? E.g., the abstract states "Using a Bayesian geostatistical framework to model travel and infection prevalence, we found off-island travel was more prevalent in and around Malabo" but I would think this should have been demonstrable from the survey data without the modeling? Explanation of what is gained through this additional effort would be useful.

2) A challenge with measuring mobility via household surveys may be that some of the most mobile individuals will not be home to answer the survey. If those responding to the survey differ meaningfully from those not present to respond, might the observed travel patterns and thus conclusions be meaningfully different?

3) Prevalence is assessed by RDT, which will not identify low density infections. Does the assumption that $r=200$ days assume that the infections are detectable by RDT throughout this period, and is that a reasonable assumption? If imported infections are actually RDT detectable for shorter periods (e.g., closer to the time of clinical episode) would this effect not potentially shift results towards a lesser contribution of imported cases to the observed prevalence?

4) As the authors know, one of the primary reasons to assess importation rates is to understand whether elimination of transmission in a given location can be a sustainable achievement. Given that Malabo appears to receive substantial importation from the mainland, and Malabo appears to be well linked by movement to other places on the island where receptivity appears to still be high, is there any way to assess the transitive importation risk into those receptive areas once elimination is achieved?

5) Mention is made of EIR measurements at sentinel sites... can any of those data be used to validate that transmission risk is indeed particularly high in places where this analysis indicates residual transmission likely remains a primary driver of observed prevalence?

Some additional minor points:

6) Fig 1 – it is hard to see much variation in the composite satellite image, nor can I see the location of the nature reserves referred to in the legend – perhaps a classified landcover map or even elevation layer might be more useful.

7) Intro: "After the completion of the current phase by the end of 2018, the project will enter a new era of malaria elimination on the island". It's not clear what this refers to, and the citation is a government report not publicly accessible. Can a brief summation be made of what this new era involves?

8) Results: "Overall, 20.2% of respondents had pernoctated away from home..." While I appreciate learning the word "pernoctate", it sounds like it means the 20% simply reported sleeping anywhere other than their home, though the methods clarify it in fact is referring to people sleeping outside their district.

Overall, a well constructed analysis that should be of substantial interest to others in our field.

Best,
Justin M. Cohen

Reviewer #1 (Remarks to the Author):

“Human mobility patterns and malaria importation on Bioko Island” is a really interesting paper and a very good case study since it is carried out in a close system (an island).

The manuscript is well written and the mathematical and statistical models well described. It lacks in showing the full results (i.e. uncertainties, variable selection, and parameterisation) and in the description of some choices and assumptions. In my view, this paper is suitable for Nature Communications but still needs essential improvements.

Major points to be addressed.

1) My major concern is related to how the data is treated. It is not clear if the authors have considered and how they considered multiple travels, i.e. same people doing within- and off-island travels in the same period. I can't see anything in the statistical and mathematical models accounting for joint (within and off in they different flavours) travels. Until this issue is not addressed, statements such as “our findings show clear human mobility patterns operating on the island and characterising them is a stepping stone towards understanding malaria importation” or “our findings suggest that malaria cases are actively imported by off-island travellers to Rio Muni, similar to what has been described in Zanzibar” are potentially highly speculative.

Responses:

This was an excellent point, and one that we had neglected to fully explain in the main text. In our response, we will first clarify what information is available in the data, and next we will explain what motivated our analyses.

The questions regarding within and off-island travel were treated separately in the MIS questionnaires; *i.e.* the same individual may have answered 'yes' to within and to off-island travel in the same 8-week period. Consequently, we treated travels to destinations within and off the island during the same period by the same people separately. That is, if the same individual travelled both to the mainland and to another district within Bioko during the 8-week period, this was counted as two different occurrences reflected in the maps of Figure 2.

In the fifth paragraph of the Discussion, we had already mentioned the following: “It was not possible to infer frequency of travel or to capture any seasonal component of either human infection or movement patterns.” To address the reviewer's comment we have expanded on this by adding the following text to the first paragraph of the Methods section, following the text: “We defined TP per *map-area* as the percentage of people in each sampling unit answering yes to within island travel, off-island travel, and both”:

“Within and off-island travel questions were treated separately in the questionnaires; that is to say, the same individual could have answered positively to history of travel off the island and within the island. It was not possible to assess frequency of travel with accuracy because the questionnaire responses were binned into three categories (1-3, 4-10 and > 10 travel occurrences within the stipulated period) with most people (97-99%, depending on the survey year) falling in the first category.”

The pattern in human mobility for residents of Bioko Island was illustrated in Figure 2 in both the raw data and the post-analysis map. The statement that malaria cases are actively imported by off-island travelers to Rio Muni was illustrated by having prevalence almost 3 times the baseline as reported in line five of Table 3. We also noted that prevalence in those traveling within the island (6.6%) was lower than the overall prevalence in non-travelers (8.8%). This was why we focused on estimating the fraction of all prevalence that could possibly be explained by travel off-island. To make it clear, we restructured the Results slightly to have three sections, roughly corresponding to the analysis of the travel data, analysis of the PfPR data, and estimation of importation. To headline the new section 1.3, we have added some text to motivate the analyses that followed.

2) Please add a table with the variable selection results and explain why you choose a VIF threshold of 10.

We have added the following text to the fifth paragraph of section 3.2 in the Methods to explain the choice of a VIF threshold of 10:

“A VIF threshold of 10 was chosen based on the commonly recommended rule of thumb for reducing collinearity. Besides, we found that a smaller or larger VIF threshold value did not influence the outcome of covariate selection.”

There were three separate analyses using some subset of variables shown in Table 4, corresponding to the three maps in the bottom row of Figure 5. In the supplement, we added tables of the regression coefficients of the final list of covariates used for each map to also address comment 4 below.

3) Reducing DIC and VIF are not the same thing, and actually reducing VIF does not imply a reduction in DIC. Can you provide more details and your assumptions about model reduction and best fitting please?

Yes, indeed, reducing the VIF was associated with reducing the degree of multi-collinearity of the covariates in a regression model and it certainly did not imply a reduction in deviance information criterion (DIC). The VIF reduction step was done prior to fitting the Bayesian geostatistical model to the data. The DIC reduction step was part of the Bayesian geostatistical model fitting and selection procedures for identifying the best-fitted and most parsimonious model that was then used to predict prevalence rates across Bioko Island. The DIC has been widely justified and recommended as a standard criterion for comparing complex Bayesian hierarchical models. We have included the above explanation in paragraph section 3.2 of the Methods:

“Subsequently, variable selection was performed using bidirectional elimination by running stepwise regressions on all model combinations using the shortlisted covariates and calculating the deviance information criterion (DIC) for each model. Finally, the best-fitted (final) model was the one with the smallest DIC value. We note that reducing the VIF was associated with reducing the degree of multi-collinearity of the covariates in a regression model and it certainly did not imply a reduction in DIC. The VIF reduction step was done prior to fitting the Bayesian geostatistical model to the data. On the other hand, the DIC reduction step was part of the Bayesian geostatistical model fitting and selection procedures for identifying the best-fitted and most parsimonious model that was then used to predict prevalence rates across Bioko Island. The DIC has been widely justified and recommended as a standard criterion for comparing complex Bayesian hierarchical models.”

4) Include tables with the Bayesian geostatistical models results, i.e. credible intervals for parameters; and the uncertainty maps.

We have added a section in the supplementary information (Section 4) in which we present the uncertainty maps according to the 95% credible intervals for predicted *PfPR* and TP. In addition, for each model we have included tables of the selected predictors and their regression coefficients.

Minor points

5) In the background section, “historically, entomological inoculation rate...”, please can you be more specific about the time frame considered in the word “historically”.

We appreciate this comment and have made the sentence more explicit by replacing it with the following and moving it forward one sentence in the same paragraph after the introduction of the BIMCP:

“Prior to this year, entomological inoculation rates (EIR) on Bioko were among the highest ever recorded for any malaria endemic area”

6) In the background section, “the project will enter a new era of malaria elimination...”, although the wording is very exciting it does not help me to understand what is this “new era” in malaria elimination.

We have reworded this sentence for more clarity:

“After the completion of the current phase by the end of 2018, strategies for the following five-year phase will aim at malaria elimination on the island.”

7) Figure 1. Include the label for the Pico Basile National Park and the Caldera de Luba Scientific research.

We have redone this figure to include both nature reserves and replacing the satellite image with a schematic map of Bioko.

8) In the results section, please can you add the proportion of total population represented by the sample size (i.e. for 2015, 17016 people are % of total population).

In Table 1, for each survey we have included the percent of the total population sampled in reference to an island-wide population census from 2015 that estimated a population of 252,000 for Bioko. We have also amended the caption to this table as well as added the following text to the first line of the first paragraph of the Results (Section 1.1): “, each year representing around 6% of the population of the island”

9) “because we did not find a significant difference between PfPr in children and adults...” I think this need to be discussed in more details, does it mean that they have similar travel and malaria PfPR patterns? If yes why?

The logic behind using PR in young children (2-10 years of age) for assessing transmission intensity in high transmission areas derives from the effect of age-acquired immunity reducing the detectability of patent infections in older children and adults. Whilst in some areas of Bioko local transmission may still result in significant levels of immunity, in many other areas transmission intensity may be too low to drive immunity. There was no significant difference between the median PfPR in all ages and in children. We removed the line cited by the reviewer as we acknowledge it can cause confusion. In replacement, we added the following text to the first paragraph of the Methods: “For the purposes of measuring malaria, we included data from individuals of all ages because we found no significant differences in PfPR between the overall sample population and children (2-10 year olds). “

Reviewer #2 (Remarks to the Author):

I enjoyed reviewing this paper, the key finding of which is that malaria prevalence in much of Bioko (particularly the main city) is consistent with being largely due to importation from the mainland, while malaria outside the main city cannot be explained by importation alone. This is an important conclusion that has critical implications for designing appropriate elimination programs in Bioko – i.e., not all places with high prevalence should necessarily be targeted with vector control measures if transmission is in fact happening elsewhere - and the methods used for deriving it are novel and potentially of use to many other countries facing similar challenges.

My chief comments and criticisms are:

1) The population surveys provided information on malaria testing and travel patterns in 185/194 map areas. Geostatistical modeling is used in this analysis to derive estimates for the entire population. These modeled extrapolations seem useful to explain, for example, what percent of the total population lives in areas defined by certain endemicity or travel characteristics, accounting for any gaps in sampling. However, it is not clear to me why the survey data could not be directly analyzed instead of using the geostatistical model results to investigate the degree to which importation drives prevalence. My concern here is that the modelling procedure is likely to smooth the data, potentially averaging over interesting heterogeneities that could hold importance for the questions posed here (i.e., pixels home to clusters of unusually high or low travel or malaria may have important roles in driving transmission). Would results change if survey-measured values were used directly to inform the model? E.g., the abstract states “Using a Bayesian geostatistical framework to model travel and infection prevalence, we found off-island travel was more prevalent in and around Malabo” but I would think this should have been demonstrable from the survey data without the modeling? Explanation of what is gained through this additional effort would be useful.

Responses:

We are grateful for the very constructive comments. Justin’s point here – that some of the variance may be real – is one that we would like to address in future studies, but we felt more comfortable using the smoothed values for this analysis. We had deliberately decided to conduct our analysis on the smoothed data precisely because sample sizes were often small, and we were concerned about making spurious conclusions.

Though data were available for 185/194 map-areas, there were several more map-areas where there were no off-island travelers (as shown in Figure 2B) translating into no denominator for estimating *PfPR* in travelers (as shown in Figure 5B), hence rendering missing values that would be reflected in the travel fraction and local residual transmission estimates. In addition, the sample size variance across map-areas was very important and in some of these mapping units sample sizes were extremely small, hence confounding any real heterogeneities (i.e. many high value pixels in Figures 2 and 5, maps A, B and C, were mostly explained by small sample sizes). Therefore, the rationale behind the use of geostatistical maps was twofold: 1) predict *PfPR* and TP in pixels where there were no data; 2) smooth out sampling variance.

To see if the answers would be different, we reran our analysis on the raw values. The answers were not substantially different.

We have replaced the text of the first paragraph of section 3.2 of the Methods with the following:

“Even though individuals were surveyed in most *map-areas*, in some, sample size variance affected the credibility of *PfPR* and TP estimates derived from the data. Moreover, there were several map-areas where there were no off-island travelers and hence no denominator for estimating *PfPR* in travelers. We therefore resorted to using the geo-positioned *PfPR* and TP raw estimates as inputs for a Bayesian geostatistical modeling framework. This methodology allowed not only to estimate *PfPR* and TP in pixels where no individuals had been sampled in order to produce continuous surfaces for each variable but also to account for the sample size variance across all *map-areas*. For *PfPR*, environmental and socio-demographic variables known to interact with and influence *PfPR* were assembled as 30 arcsecond spatial grids. For predicting TP, the covariates used were population density and distance to Malabo (Table 4). These variables were also input data for the Bayesian geostatistical model.”

We added the following text to the caption of Figures 2 and 5: “The noise introduced by sampling variance, illustrated by the extreme values of the top row, was smoothed out by the geostatistical models (bottom row).” We also have modified the caption of Figure 5 to more clearly explain what are the missing data referred to above and have added the local residual transmission and travel fraction maps derived from the raw data to Figure 6. We edited the following text to Section 1.3: “Figure 6 plots the travel fraction and local residual transmission estimates by map-area based on the raw data and on the predicted surfaces. The former suffered from noisy data and several missing values”. The caption to Figure 6 has also been updated to incorporate these two new maps.

2) A challenge with measuring mobility via household surveys may be that some of the most mobile individuals will not be home to answer the survey. If those responding to the survey differ meaningfully from those not present to respond, might the observed travel patterns and thus conclusions be meaningfully different?

This is one limitation of the data that we did not sufficiently clarify in the caveats described in the Discussion. Working adult men are known to be often not present during the surveys and may travel more than the rest of the population. The extent of this bias was not possible to assess from the data, however. We have added the following line to the fifth paragraph of the Discussion:

“Third, the MIS sampling is sometimes biased by the absence of working adult men during the surveys; since this cohort of the population may have different travel habits, this introduced an unknown bias into our estimates.”

3) Prevalence is assessed by RDT, which will not identify low density infections. Does the assumption that $r=200$ days assume that the infections are detectable by RDT throughout this period, and is that a reasonable assumption? If imported infections are actually RDT detectable for shorter periods (e.g., closer to the time of clinical episode) would this effect not potentially shift results towards a lesser contribution of imported cases to the observed prevalence?

We agree with the reviewer in that RDT sensitivity introduces yet another source of bias into our estimates. As the reviewer points out, however, if imported infections were detectable by RDT for shorter periods of time this would effectively reduce the r parameter to some degree. The sensitivity of our model to this parameter was tested in the supplementary information (Section 3.2) by assuming half the value ($r = 100$), which showed that the travel fraction estimates indeed decreased but that the increase in the local residual transmission estimates was marginal. This change in the r parameter could also be applied to the lower sensitivity of RDTs.

We also note that the $r = 200$ days was the figure used to track the change in PfPR over time after the total interruption of transmission by light microscopy. The model seemed to fit data very well over a three year period. As far as we know, RDT results tend to be consistent with results of light microscopy. We are not aware of any data suggesting PfPR over time by RDT in a cohort of untreated infections would look different. The question of treatment, however, is a much larger problem, as it is far more common nowadays for an infection to be cleared by drugs rather than naturally. This was why our sensitivity analysis looked at shorter durations of infection.

4) As the authors know, one of the primary reasons to assess importation rates is to understand whether elimination of transmission in a given location can be a sustainable achievement. Given that Malabo appears to receive substantial importation from the mainland, and Malabo appears to be well linked by movement to other places on the island where receptivity appears to still be high, is there any way to assess the transitive importation risk into those receptive areas once elimination is achieved?

There is a way to assess the rate of importations to receptive areas within a simulation modeling framework, which we plan to describe in a forthcoming publication. The simulation model would include as an input the amount of time spent exposed in different locations; for this we would need to know how long infectious people visit elsewhere on the island, which is a question that was suggested for the new MIS questionnaire. These new data will be integrated within ongoing modeling work that is beyond the scope of this paper, but that is indeed being prepared for a future manuscript.

5) Mention is made of EIR measurements at sentinel sites... can any of those data be used to validate that

transmission risk is indeed particularly high in places where this analysis indicates residual transmission likely remains a primary driver of observed prevalence?

Unfortunately, EIR has been measured only in a handful of sentinel sites ($n=11 \times 3 \text{ years} = 33$). It would be difficult to make any meaningful validation of local residual transmission estimates using these data. With that said, we observed that all EIR estimates above the 3rd quartile of their distribution, or 15.4 ibppy, were recorded in areas with $LRT > 14.7\%$ (see plot below). We note that we do plan to include analysis of the mosquito studies in the new manuscript describing a simulation model for the island.

For clarity, we added the following text to the third paragraph of the Discussion: “Unfortunately, given the limited number of such estimates and the disproportionately smaller number of sites located in urban Malabo, these data proved insufficient to cross-validate our estimates of local residual transmission.”

Some additional minor points:

6) Fig 1 – it is hard to see much variation in the composite satellite image, nor can I see the location of the nature reserves referred to in the legend – perhaps a classified landcover map or even elevation layer might be more useful.

Figure 1 has been updated to make it clearer based on comments from both reviewers.

7) Intro: “After the completion of the current phase by the end of 2018, the project will enter a new era of malaria elimination on the island”. It’s not clear what this refers to, and the citation is a government report not publicly accessible. Can a brief summation be made of what this new era involves?

We have added the following text to explain this further:

“After the completion of the current phase by the end of 2018, strategies for the following five-year phase will aim at malaria elimination on the island. Whilst the specific plans are still being delineated, the inclusion of a pre-erythrocytic vaccine to the arsenal of control interventions is under consideration.”

8) Results: “Overall, 20.2% of respondents had pernoctated away from home...” While I appreciate learning the word “pernoctate”, it sounds like it means the 20% simply reported sleeping anywhere other than their home, though the methods clarify it in fact is referring to people sleeping outside their district.

We have clarified this sentence by adding:

“Overall, 20.2% of respondents had pernoctated at either another district or outside of Bioko at least one night within the preceding eight weeks;”

Overall, a well constructed analysis that should be of substantial interest to others in our field.

Best,
Justin M. Cohen

REVIEWERS' COMMENTS:

Reviewer #2 (Remarks to the Author):

I appreciate the responses to my questions and feel the changes that have been made satisfactorily address them.

Reviewers' comments:

Reviewer #3 (Remarks to the Author):

NCOMMS-18-32124A

Title *Human Mobility Patterns and Malaria Importation on Bioko Island*

Task: "if you could review the authors' revisions in response to the concerns raised by Reviewer 1. We would ask you not to raise completely new points in the review at this stage, unless of course you feel that leaving these unaddressed should preclude publication"

Overall, I think that the authors have adequately addressed the comments raised by Reviewer 1. I have added specific comments under each response (in *italic*).

Note: I did not have access to the original manuscript submitted or the files with the comments raised by the reviewers. Therefore, my review is based on the track-change version (pdf with edited sections highlighted in yellow) and the response to the reviewers file.

Reviewer #1 (Remarks to the Author):

R1: "Human mobility patterns and malaria importation on Bioko Island" is a really interesting paper and a very good case study since it is carried out in a close system (an island). The manuscript is well written and the mathematical and statistical models well described. It lacks in showing the full results (i.e. uncertainties, variable selection, and parameterisation) and in the description of some choices and assumptions. In my view, this paper is suitable for Nature Communications but still needs essential improvements.

Major points to be addressed.

1) My major concern is related to how the data is treated. It is not clear if the authors have considered and how they considered **multiple travels**, i.e. same people doing within- and off-island travels in the same period. I can't see anything in the statistical and mathematical models accounting for joint (within and off in they different flavours) travels. Until this issue is not addressed, statements such as "our findings show clear human mobility patterns operating on the island and characterising them is a stepping stone towards understanding malaria importation" or "our findings suggest that malaria cases are actively imported by offisland travellers to Rio Muni, similar to what has been described in Zanzibar" are potentially highly speculative.

Responses:

This was an excellent point, and one that we had neglected to fully explain in the main text. In our response, we will first clarify what information is available in the data, and next we will explain what motivated our analyses.

The questions regarding within and off-island travel were treated separately in the MIS questionnaires; i.e. the same individual may have answered 'yes' to within and to off-island travel in the same 8-week period. Consequently, we treated travels to destinations within and off the island during the same period by the same people separately. That is, if the same individual travelled both to the mainland and to another district within Bioko during the 8-week period, this was counted as two different occurrences reflected in the maps of Figure 2.

In the fifth paragraph of the Discussion, we had already mentioned the following: "It was not possible to infer frequency of travel or to capture any seasonal component of either human infection or movement patterns."

To address the reviewer's comment we have expanded on this by adding the following text to the first paragraph of the Methods section, following the text: "We defined TP per map-area as the percentage of people in each sampling unit answering yes to within island travel, off-island travel, and both":

"Within and off-island travel questions were treated separately in the questionnaires; that is to say, the same individual could have answered positively to history of travel off the island and within the island. It was not possible to assess frequency of travel with accuracy because the questionnaire responses were binned into three categories (1-3, 4-10 and > 10 travel occurrences within the stipulated period) with most people (97-99%, depending on the survey year) falling in the first category."

The pattern in human mobility for residents of Bioko Island was illustrated in Figure 2 in both the raw data and the post-analysis map. The statement that malaria cases are actively imported by off-island travelers to Rio Muni was illustrated by having prevalence almost 3 times the baseline as reported in line five of Table 3. We also noted that prevalence in those traveling within the island (6.6%) was lower than the overall prevalence in non-travelers (8.8%). This was why we focused on estimating the fraction of all prevalence that could possibly be explained by travel off-island. To make it clear, we restructured the Results slightly to have three sections, roughly corresponding to the analysis of the travel data, analysis of the PfPR data, and

estimation of importation. To headline the new section 1.3, we have added some text to motivate the analyses that followed.

- *The authors have adequately addressed R1's question on travel frequency by explaining how in-land and off-island have been distinguished and treated in the analysis and why the data did not allow to further stratify the analysis using the number of travels (97-99%, depending on the survey year) falling in the first category).*

2) Please add a table with the variable selection results and explain why you choose a VIF threshold of 10.

We have added the following text to the fifth paragraph of section 3.2 in the Methods to explain the choice of a VIF threshold of 10:

"A VIF threshold of 10 was chosen based on the commonly recommended rule of thumb for reducing collinearity. Besides, we found that a smaller or larger VIF threshold value did not influence the outcome of covariate selection."

There were three separate analyses using some subset of variables shown in Table 4, corresponding to the three maps in the bottom row of Figure 5. In the supplement, we added tables of the regression coefficients of the final list of covariates used for each map to also address comment 4 below.

- *The authors have added useful explanation in section 3.2 of the Methods. I believe that the tables added in the supplement sufficiently address this point.*

3) Reducing DIC and VIF are not the same thing, and actually reducing VIF does not imply a reduction in DIC. Can you provide more details and your assumptions about model reduction and best fitting please?

Yes, indeed, reducing the VIF was associated with reducing the degree of multi-collinearity of the covariates in a regression model and it certainly did not imply a reduction in deviance information criterion (DIC). The VIF reduction step was done prior to fitting the Bayesian geostatistical model to the data. The DIC reduction step was part of the Bayesian geostatistical model fitting and selection procedures for identifying the best fitted and most parsimonious model that was then used to predict prevalence rates across Bioko Island. The DIC has been widely justified and recommended as a standard criterion for comparing complex Bayesian hierarchical models. We have included the above explanation in paragraph section 3.2 of the Methods: "Subsequently, variable selection was performed using bidirectional elimination by running stepwise regressions on all model combinations using the shortlisted covariates and calculating the deviance information criterion (DIC) for each model. Finally, the best-fitted (final) model was the one with the smallest DIC value. We note that reducing the VIF was associated with reducing the degree of multi-collinearity of the covariates in a regression model and it certainly did not imply a reduction in DIC. The VIF reduction step was done prior to fitting the Bayesian geostatistical model to the data. On the other hand, the DIC reduction step was part of the Bayesian geostatistical model fitting and selection procedures for identifying the best fitted and most parsimonious model that was then used to predict prevalence rates across Bioko Island. The DIC has been widely justified and recommended as a standard criterion for comparing complex Bayesian hierarchical models."

- *The authors acknowledge R1's criticism and explain the different role of reducing VIF and reducing DIC in model selection. Because the whole model selection procedure has been based on both VIF and DIC, I would suggest not to use the term "best-fitting model" (I think R1 was hinting at that) but rather "final model", or something similar.*

4) Include tables with the Bayesian geostatistical models results, i.e. credible intervals for parameters; and the uncertainty maps.

We have added a section in the supplementary information (Section 4) in which we present the uncertainty maps according to the 95% credible intervals for predicted PfPR and TP. In addition, for each model we have included tables of the selected predictors and their regression coefficients.

- *Tables and uncertainty maps have been added.*

Minor points

5) In the background section, "historically, entomological inoculation rate...", please can you be more specific about the time frame considered in the word "historically".

We appreciate this comment and have made the sentence more explicit by replacing it with the following and

moving it forward one sentence in the same paragraph after the introduction of the BIMCP:
“Prior to this year, entomological inoculation rates (EIR) on Bioko were among the highest ever recorded for any malaria endemic area”

➤ *Comment appropriately addressed.*

6) In the background section, “the project will enter a new era of malaria elimination...”, although the wording is very exciting it does not help me to understand what is this “new era” in malaria elimination.

We have reworded this sentence for more clarity:

“After the completion of the current phase by the end of 2018, strategies for the following five-year phase will aim at malaria elimination on the island.”

➤ *The authors have rephrased the sentence following R1’s comment.*

7) Figure 1. Include the label for the Pico Basile National Park and the Caldera de Luba Scientific research.

We have redone this figure to include both nature reserves and replacing the satellite image with a schematic map of Bioko.

➤ *The map in Figure 1 include Pico Basile and Caldera de Luba (I have no access to the previous map to be able to compare)*

8) In the results section, please can you add the proportion of total population represented by the sample size (i.e. for 2015, 17016 people are % of total population).

In Table 1, for each survey we have included the percent of the total population sampled in reference to an island-wide population census from 2015 that estimated a population of 252,000 for Bioko. We have also amended the caption to this table as well as added the following text to the first line of the first paragraph of the Results (Section 1.1): “, each year representing around 6% of the population of the island”

➤ *Proportions are in Table 1.*

9) “because we did not find a significant difference between PfPr in children and adults...” I think this need to be discussed in more details, does it mean that they have similar travel and malaria PfPR patterns? If yes why?

The logic behind using PR in young children (2-10 years of age) for assessing transmission intensity in high transmission areas derives from the effect of age-acquired immunity reducing the detectability of patent infections in older children and adults. Whilst in some areas of Bioko local transmission may still result in significant levels of immunity, in many other areas transmission intensity may be too low to drive immunity. There was no significant difference between the median PfPR in all ages and in children. We removed the line cited by the reviewer as we acknowledge it can cause confusion. In replacement, we added the following text to the first paragraph of the Methods: “For the purposes of measuring malaria, we included data from individuals of all ages because we found no significant differences in PfPR between the overall sample population and children (2-10 year olds). “

➤ *The authors added a comment to address R1’s concern.*

26 April 2019

To the referees,

Below please find our responses to your comments of our revised manuscript entitled "HUMAN MOBILITY PATTERNS AND MALARIA IMPORTATION ON BIOKO ISLAND" (NCOMMS-18-32124A).

We have addressed the following request by Reviewer #3 in relation to Major point 3 from Reviewer #1:

R3: The authors acknowledge R1's criticism and explain the different role of reducing VIF and reducing DIC in model selection. Because the whole model selection procedure has been based on both VIF and DIC, I would suggest not to use the term "best-fitting model" (I think R1 was hinting at that) but rather "final model", or something similar.

We replaced all occurrences of "best-fitted model" with "final model" within the Methods section of our manuscript. Reviewer #3 also stated that all other comments by Reviewer #1 had been addressed satisfactorily.

Reviewer #2 expressed: "I appreciate the responses to my questions and feel the changes that have been made satisfactorily address them", hence we did not need addressing any further issues.

Once again, we are grateful to all three reviewers for their constructive feedback, which we feel helped improve the scientific quality of our manuscript.

Sincerely,

Dr. Carlos A Guerra
On behalf of all authors